# Anti-Inflammation and Anti-Pyroptosis Activities of Mangiferin via Suppressing NF-κB/NLRP3/GSDMD Signaling Cascades

**DOI:** 10.3390/ijms231710124

**Published:** 2022-09-04

**Authors:** Min Feng, Shaoqiang Wei, Shidong Zhang, Ying Yang

**Affiliations:** 1College of Veterinary Medicine, Inner Mongolia Agricultural University, Hohhot 010018, China; 2Engineering Technology Research Center of Traditional Chinese Veterinary Medicine of Gansu Province, Lanzhou Institute of Animal Husbandry and Pharmaceutical Sciences of Chinese Academy of Agricultural Sciences, Lanzhou 730050, China

**Keywords:** mangiferin, LPS, pyroptosis, inflammatory caspases, GSDMD

## Abstract

Mangiferin (MF), a xanthone that extensively exists in many herbal medicines, processes significant activities of anti-inflammation and immunomodulation. The potential regulatory effect and mechanism of mangiferin on cell pyroptosis remain unclear. In this study, mouse bone-marrow-derived macrophages (BMDMs) were stimulated with 1 μg/mL LPS to induce cell pyroptosis and were treated with 10, 50, or 100 μg/mL MF for regulating pyroptosis. The cell supernatants TNF-α, IL-1β, IL-6, and IL-18 were detected by enzyme-linked immunosorbent assay (ELISA); gene expression of *TNF-α*, *IL-1β*, *IL-6*, *IL-18*, *Caspase-1*, *Caspase-11*, and *gasdermin D* (*GSDMD*) was tested by real-time polymerase chain reaction (RT-PCR), and protein expression levels of apoptosis-associated speck-like protein containing a caspase-recruitment domain (ASC), nod-like receptor protein-3 (NLRP3), caspase-1, caspase-11, GSDMD, and NF-κB were detected by Western blot. The results showed that MF significantly inhibited the secretion and gene expression of TNF-α, IL-6, IL-1β, and IL-18 that were elevated by LPS. Moreover, MF significantly suppressed the gene expression of *Caspase-1*, *Caspase-11*, and *GSDMD*, and decreased the protein levels of NLRP3, caspase-1, caspase-11, full-length GSDMD (GSDMD-FL), GSDMD N-terminal (GSDMD-N), and NF-κB. In conclusion, mangiferin has a multi-target regulating effect on inflammation and pyroptosis by inhibiting the NF-κB pathway, suppressing inflammatory caspase-mediated pyroptosis cascades, and reducing GSDMD cleavage in LPS-induced BMDMs.

## 1. Introduction

Pyroptosis is a type of programmed cell death in which cells swell until their membranes rupture, resulting in the release of cell contents that activate a strong inflammatory response [1]. Pyroptosis is an important innate immune response and plays an important role in fighting infection [1,2]. However, excessive pyroptosis is an essential factor for Gram-negative bacteria to cause severe infection [3]. Pyroptosis is characterized by activation of nod-like receptor protein-3 (NLRP3) inflammasome, inflammatory caspase activation, and the release of IL-1β and IL-18 [4,5,6]. It is the main reason for the suppression and failure of immune function in infective animals, and solving excessive pyroptosis to get rid of immunosuppression may be one of the fundamental breakthroughs in the treatment of heavy infection [7].

Immune cell pyroptosis, such as macrophages and dendritic cells, is an important factor in sepsis that causes the body’s immune response to be out of control [2]. While macrophages are the most plastic cells acting as the main and primary barrier in various tissues of the body against foreign invaders [8], massive stimulation of macrophages can lead to fatal septic shock syndrome and multiple organ failure [9,10,11]. The literature is replete with reports that LPS can induce pyroptosis of macrophages by activating caspase-1-mediated canonical inflammasome [12] or binding to its intracellular receptor caspase-4/5/11 to cleave gasdermin D to produce N-terminal and execute pyroptosis [13]. 

With the increasing incidence of antibiotic resistance, traditional Chinese medicine has shown more and more advantages in the treatment of bacterial sepsis [14,15]. Mangiferin is a xanthone (Figure 1) that extensively exists in many higher plants, especially in many herbal medicines [16]. Pharmacological studies indicate that mangiferin is related to anti-inflammatory, antipyretic, analgesic, antibacterial, immune regulation, hypoglycemic, and antioxidant effects [17]. For example, mangiferin significantly reduced NO, IL-1β, IL-6, and TNF-α by inhibiting activation of NF-κB in LPS-induced mouse microglia cells [18], and decreased IL-1β, IL-6, and TNF-α by inhibiting the nuclear translocation of the NF-κB p65 subunit in mononuclear macrophages, and protected mouse lung injury against sepsis [19]. Meanwhile, mangiferin may interact with TLR4/PI3K/AKT/NFκB signaling to inhibit NLRP3 inflammasome activation and modulate GSDMD-mediated pyroptosis in rats [20], and it alleviated the histopathology of inflammation and the levels of pro-inflammatory cytokines (TNF-α, IL-1β, and IL-6) via inhibiting NF-κB and NLRP3 inflammasome activation in lactating female mice [21]. Therefore, mangiferin has significant pharmacological function in inhibiting inflammation and inhibiting pyroptosis in both cell models and mammalian models. 

Given the significant activities of anti-inflammation response and anti-pyroptosis of mangiferin, the mechanism of mangiferin in regulating pyroptosis and inflammation remains in need of further study. In this study, we explored the function and mechanism of mangiferin against pyroptosis in bone-marrow-derived macrophages (BMDMs) induced by LPS, which was an inflammation model of progenitor macrophages. The results showed that mangiferin can inhibit the cleavage of GSDMD and the activation of inflammatory caspase-1 and 11, which demonstrated the pharmacological targets for drug development. 

## 2. Results

### 2.1. Cytotoxicity of Mangiferin in BMDMs

The cells were treated with different concentrations of mangiferin for 24 h, and cell viability was detected by CCK8. The result showed that cell-proliferation-inhibiting rates were increased in a linear manner along with the increasing of drug concentrations (Figure 2). Based on the equation of linear regression y = 0.1549x − 2.9229 (R² = 0.9887), IC_50_ was calculated as 352.80 μg/mL. Therefore, when the mangiferin was lower than 352.80 µg/mL, it had no obvious cytotoxic effect on BMDMs.

### 2.2. Mangiferin Inhibits Inflammatory Cytokines

As shown in Figure 3, compared with the negative control (CN), the cell supernatant cytokines of IL-1β, TNF-α, IL-6, and IL-18 were significantly increased by LPS. After treatment with mangiferin, compared with LPS groups, the secretion levels of IL-1β, TNF-α, and IL-6 were significantly inhibited by the drug at doses of 10, 50, and 100 μg/mL (*p* < 0.05), whereas IL-18 was just decreased by mangiferin at the dose of 100 μg/mL (*p* < 0.05). Thus, mangiferin suppresses the secretion of pro-inflammatory cytokines in BMDMs induced by LPS.

Figure 4 shows the effect of mangiferin on gene expression of pro-inflammatory cytokines. Compared with the CN group, the gene expression changes of *TNF-α*, *IL-1β*, *IL-6*, and *IL-18* were significantly increased by LPS. After mangiferin treatment, the above gene expression levels were significantly downregulated (*p* < 0.05), which was in a dose-dependent manner. Thus, mangiferin inhibits the gene expression of pro-inflammatory cytokines that were increased by LPS in BMDMs.

### 2.3. Mangiferin Reduces Cell Pyroptosis Morphology

It can be observed from Figure 5 that normal BMDMs are mostly round or elliptical, with short protrusions, and those with active functions often protrude longer pseudopods, showing irregular shapes. The nucleus is small, round or oval, and darker in color. After treatment with 1 μg/mL LPS for 24 h, some BMDMs expanded, and many bubble-like protrusions appeared before the rupture of the plasma membrane, followed by the overall flattening of the cells at a later stage. After mangiferin treatment, the phenomenon and quantity of pyroptosis cells were significantly reduced.

Scanning electronic microscopy observed that normal BMDMs were spherical with a smooth surface (Figure 6A), whereas the pyroptotic cell showed swelling, protruding, and rupturing induced by LPS (Figure 6B). After mangiferin treatment, the morphological characteristics of pyroptosis were markedly reduced (Figure 6C).

### 2.4. Mangiferin Inhibits Inflammatory Caspases in BMDMs

The results of gene expression changes of *Caspase-1*, *Caspase-11*, and *GSDMD* are shown in Figure 7. Compared with the CN group, LPS significantly increased the mRNA level of each gene in BMDMs. After mangiferin treatment, compared with the LPS group, their mRNA levels were significantly downregulated (*p* < 0.05). In addition, the inhibiting effect of mangiferin on gene expression of *Caspase-1*, *Caspase-11*, and *GSDMD* was dose-dependent.

### 2.5. Mangiferin Inhibits Pyroptotic Cascades in BMDMs

Figure 8 shows that the protein expression levels of ASC, NLRP3, CASP1, CASP11, GSDMD-FL (full-length GSDMD), GSDMD-N (GSDMD N-terminal), and NF-κB were markedly higher than that in the CN group, which was induced by LPS. Compared with the LPS group, the elevated protein levels were partly reduced after the treatment with mangiferin. Specifically, MF significantly decreased NLRP3, CASP1, and ASC, which were critical members of classical inflammasome, while ASC failed to be drastically changed. GSDMD-FL and GSDMD-N were reduced with treatment of MF, but seem to be independent of drug concentrations. Moreover, NF-κB (p65) was significantly inhibited or inactivated in a drug-concentration-dependent manner. 

Based on the protein band gray levels analysis, the statistical results (Figure 9) indicated that mangiferin significantly suppressed the protein levels of NLRP3, CASP1, CASP11, GSDMD-FL, and NF-κB (*p* < 0.05). While mangiferin downregulated the expression of ASC and GSDMD-N, it was independent of drug concentrations. Specifically, ASC was inhibited by 50 μg/mL mangiferin but 10 and 100 μg/mL had no inhibitory effect. Likewise, 10 and 100 μg/mL mangiferin significantly inhibited GSDMD-N but 50 μg/mL had no inhibitory effect. 

## 3. Discussion

The goal of this research was to confirm whether mangiferin inhibits LPS-induced pyroptosis in mouse bone-marrow-derived macrophages. We investigated the effect of mangiferin on LPS-induced pyroptosis in vitro, and found that mangiferin has an anti-pyroptosis function. In terms of mechanism, mangiferin inhibited expression and activation of caspase-1/11 and NLRP3, and blocked cutting of GSDMD and N-terminal domain formation as well as contributed to pyroptosis suppression. This may shed light on a new mechanism for mangiferin to fight against pyroptosis. 

Mangiferin possesses numerous pharmacological properties including antioxidative, antiaging, antitumor, antibacterial, antiviral, immunomodulatory, antidiabetic, hepatoprotective, and analgesic effects [22,23,24,25]. Recently, research studies have reported that mangiferin has important roles in anti-inflammation through the suppression of NF-κB and the MAPK signaling pathway in RAW264.7 macrophages [26] and immunomodulation via regulating the Bregs level and activating the Nrf2 antioxidant pathway [27]. Here, our results demonstrated that mangiferin significantly inhibited the release of pro-inflammatory cytokines as well as their mRNA expression, including *TNF-α*, *IL-6*, *IL-18*, and *IL-1β* (Figure 3 and Figure 4). Thus, mangiferin has significant anti-inflammatory effect on BMDMs. Moreover, it was reported that mangiferin can reduce inflammatory response by inhibiting NLRP3 inflammasome in an NF-κB-dependent manner in macrophages [28]. As NLRP3 is a critical contributor to inflammasome and canonical pyroptosis [29], to this point, the effect of mangiferin on pyroptosis was investigated through morphology study, and the results showed mangiferin also significantly suppressed the pyroptotic morphology in BMDMs (Figure 5 and Figure 6), which is characterized by nuclear shrinkage, the swell of plasma membrane, and content bursting [30]. These results directly suggest that mangiferin indeed inhibited LPS-induced inflammation and pyroptosis in BMDMs.

NLRP3 is an intracellular sensor that detects a variety of endogenous danger signals, resulting in the formation and activation of the NLRP3 inflammasome [31]. NLRP3 inflammasome is a multi-protein complex consisting of NLRP3, ASC, and caspase-1 [32]. The activation of NLRP3 inflammasome controls caspase-1 activity in the innate immune system, which leads to gasdermin-D-mediated pyroptotic cell death [30]. NLRP3, caspase-1, and ASC have been exploited as new therapeutic targets against pyroptosis and sepsis [33]. Regarding anti-pyroptosis, the inhibiting effect of mangiferin on NLRP3 inflammasome has been demonstrated [27], while caspase-1 activation, as well as GSDMD-N formation, failed to be revealed, which was the critical executor of pyroptosis through opening the cell membrane pores [34,35,36]. In the direct comparison of these before and after mangiferin treatment, it showed that mangiferin significantly inhibited gene expression of inflammatory *Caspase-1*/*11* and *GSDMD* (Figure 7), and activation of NLRP3 and caspase-1, and formation of GSDMD N-terminal (Figure 8). We suggest that mangiferin inhibited pyroptosis through suppressing both caspase-1 activation and GSDMD-N formation, and these anti-pyroptotic functions of mangiferin are clear and well documented. 

Additionally, it is well known that LPS can directly activate murine inflammatory caspase-11 to induce pyroptotic cell death that is defined as noncanonical pyroptosis [37]. Recently, research showed that caspase-11 promotes NLRP3 inflammasome activation [38], and NLRP3 inflammasome was necessary for activation of LPS-bound pro-caspase-11 [39,40]. In this study, the results showed that LPS effectively activated caspase-11 expression, and mangiferin can regulate the expression or activation of caspase-11 as it regulated caspase-1 and NLRP3 (Figure 8 and Figure 9). Therefore, mangiferin not only has the inhibitory effect on canonical pyroptosis mediated by caspase-1, but also has an inhibitory effect on noncanonical pyroptosis mediated by caspase-11. We suggest that there are multiple targets in the regulation of pyroptosis by mangiferin. 

## 4. Materials and Methods

### 4.1. Reagents and Cell Culture

Mangiferin (Must-20041123, ≥98.21%) was purchased from Must Biotechnology Co., Ltd. (Chengdu, China) and was dissolved in DMSO when used for the experiment. Dulbecco’s modified eagle medium (DMEM), phosphate buffer solution (PBS), and fetal bovine serum (FBS) were purchased from Gibco (Grand Island, NE, USA). Trypsin and lipopolysaccharide (LPS, O111: B4) were purchased from Sigma-Aldrich LLC (Shanghai, China). Cell counting kit-8 (CCK8) was purchased from Biosharp Life Science (Hefei, China). TBST, TBS, ECL (PE0010-B), ELISA kits for mouse TNF-α, IL-1β, IL-6, and IL-18 were purchased from Solarbio Biotechnology Co., Ltd. (Beijing, China). Molpure^®^ cell total RNA kit (Cat#19221) was purchased from Yeasen Biotechnology (Shanghai, China). Evo M-MLV RT-PCR Kit and SYBR^®^ Green Premix Pro Taq HS qPCR Kit were purchased from Accurate Biology (Changsha, China). ProteinExt^®^ Mammalin Total Protein Extraction was purchased from TransGen Biotech (Beijing, China). BCA protein assay kit was purchased from Takara (Dalian, China). Mouse caspase-1 antibody (GTX101322) was purchased from GeneTex Inc. (Irvine, CA, USA), mouse caspase-11 antibody (ab180673) and mouse NLRP3 antibody (ab263899) were purchased from Abcam (Boston, MA, USA). Mouse GSDMDC1 antibody (sc-393656), mouse ASC antibody (sc-514414), and mouse NF-κB p65 antibody (sc-8008) were purchased from Santa Cruz Biotechnology Inc. (Shanghai, China). Mouse β-Tubulin Monoclonal antibody (YM3030) and HRP-conjugated goat anti-mouse IgG (rs0001) were purchased from Immunoway Biotechnology (Beijing, China). 

### 4.2. Mangiferin Cytotoxicity Assay

Mouse bone-marrow-derived macrophages were grown in DMEM; all media were supplemented with 10% FBS. The cells were grown at 37 °C in a 5% CO_2_ incubator. The cells were seeded in a 96-well culture plate at a density of 2 × 10^4^ cells/well. After being cultured for 24 h, the cells were treated with different doses of mangiferin (31.25, 62.5, 125, 250, and 500 μg/mL) in sextuplicate for 24 h. We quantified the survival rate of cells by determining optical density (OD) at 450 nm using a CCK8 kit, then the cell proliferation inhibiting rate was calculated using the following formula: (OD_control_ − OD_treated_)/(OD_control_ − OD_blank_) × 100%. 

### 4.3. Cell Morphology Observation

The cells were seeded in a 6-well culture plate containing slides at a density of 2 × 10^4^ cells/well. After being cultured for 24 h, the cells were treated with LPS (1 μg/mL) for 6 h, then treated with different doses of mangiferin (10, 50, and 100 μg/mL) for 24 h. Cell morphological changes were observed under phase-contrast microscopy (Leica, DMi1, Suzhou, China) and scanning electron microscopy (Inspect, FEI, Hillsboro, OR, USA), respectively. 

### 4.4. Cytokines Assay by ELISA

Cell grouping and treatment were performed as previously described. After treatment with LPS and mangiferin, the cell supernatant cytokines of TNF-α, IL-1β, IL-6, and IL-18 were determined by ELISA kits, according to the manufacturer’s protocols, respectively. 

### 4.5. mRNA Expression Detection by Real-Time PCR

Cellular total RNA was isolated using a Molpure cell total RNA kit (Yeasen, Shanghai, China). The quantity and quality of RNA were determined with OD_260/280_ using an ultramicrospectrophotometer (Eppendorf, Hamburg, Germany), then 1 μg RNA was converted into cDNA using an Evo M-MLV RT Kit (TaKaRa, Dalian, China). Real-time PCR reaction was performed on the Bio-Rad CFX cycler using SYBR^®^ Green (Bio-Rad, San Francisco, CA, USA). PCR was performed for 40 cycles using the following conditions: pre-incubation at 95 °C for 30 s, denaturation at 95 °C for 5 s, and annealing at 60 °C for 30 s. Every reaction was conducted in triplicates. The primers for the mRNA expression test were as follows: 5′-catgtacgttgctatccaggc-3′ and 5′-ctccttaatgtcacgcacgat-3′ for *β-actin*; 5′-atccatctctttgcggaggc-3′ and 5′-gggggagaggtagggatgtt-3′ for *TNF-α*; 5′-tgccaccttttgacagtgatg-3′ and 5′-aaggtccacgggaaagacac-3′ for *IL-1β*; 5′-gccttcttgggactgatgct-3′ and 5′-gacaggtctgttgggagtgg-3′ for IL-6; 5′-cctttgaggcatccaggaca-3′ and 5′-cacaccacaggggagaagtg-3′ for *IL-18*; 5′-aggcacgggacctatgtgat-3′ and 5′-agggcaaaacttgagggtcc-3′ for *Caspase-1*; 5′-tggaacagctgggcaaagaa-3′ and 5′-gtcactgcgttcagcattgt-3′ for *Caspase-11*; 5′-gatcaaggaggtaagcggca-3′ and 5′-cactccggttctggttctgg-3′ for *GSDMD*.

### 4.6. Protein Expression Assay by Western Blotting

Western blot was used to detect protein expression of ASC, NLRP3, CASP1, CASP11, GSDMD-FL, GSDMD-N, and NF-κB in mangiferin-treated cells from BMDMs. Briefly, the cells were lysed with Total Protein Extraction buffer (Transgen, Beijing, China), and the protein concentration was determined by a BCA kit. Every total protein sample (45 µg) was separated by 8–12% SDS-polyacrylamide gel electrophoresis (SDS-PAGE). Then, proteins were transferred onto 0.45 μm polyvinylidene fluoride membrane using a semidry transfer cell (Bio-Rad, Hercules, CA, USA). The membrane was blocked with 5% nonfat milk for 1.5 h at room temperature, and then incubated with primary antibodies overnight at 4 °C. After that, the membrane was washed with TBST, then incubated with HRP-conjugated secondary antibodies for 1 h at room temperature, and washed again by TBST; lastly, protein bands were visualized using ECL and photographed. Image J software was used to analyze the gray value of the protein bands.

### 4.7. Statistical Analyses

The relative quantification of target genes was determined by calculating the ratio between the target gene and *β-actin* using the method of 2^−ΔΔCt^. The scoring of data was performed using GraphPad Prism (v8.0.1, GraphPad Software, San Diego, CA, USA); normally distributed data from different groups were compared by one-way ANOVA with *p* < 0.05 being considered statistically significant.

## 5. Conclusions

This research, based on the anti-inflammation response and anti-pyroptosis activities of mangiferin, demonstrated that mangiferin has a multi-target regulating function on inflammation and pyroptosis by inhibiting the caspase-1/11 pathway, suppressing NF-κB/NLRP3-mediated pyroptosis cascades, and reducing GSDMD cleavage in LPS-induced mouse bone-marrow-derived macrophages. While mangiferin has multi-target pharmacological activity, it warrants further studies on the regulating function and mechanism of mangiferin in LPS pyroptosis. In addition, the anti-pyroptosis roles of mangiferin should also be further investigated in other cells.

## Figures and Tables

**Figure 1 ijms-23-10124-f001:**
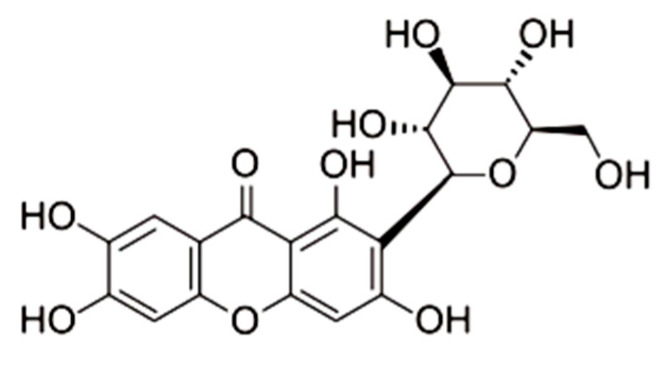
Molecular structure of mangiferin.

**Figure 2 ijms-23-10124-f002:**
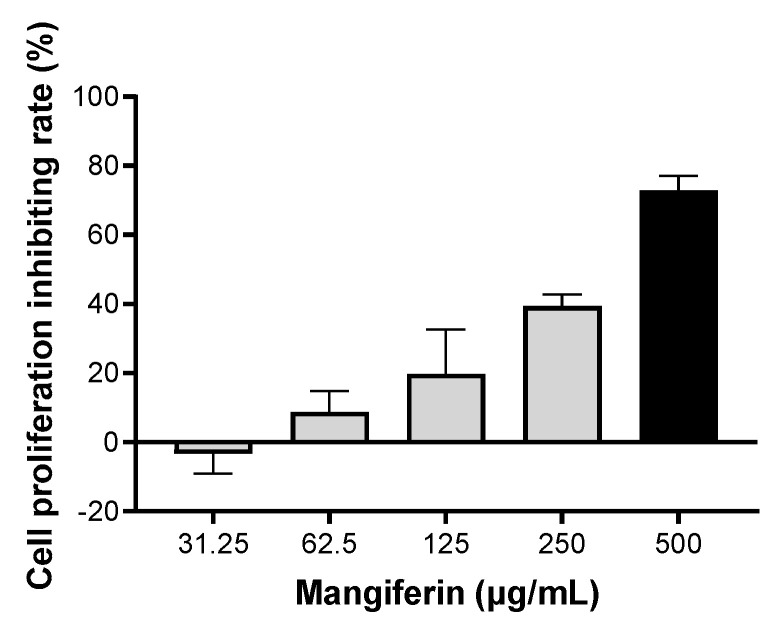
Cell proliferation inhibiting rate of mangiferin in BMDMs. The cells were seeded in a 96-well culture plate and cultured for 24 h, then the cells were treated with different doses of mangiferin for another 24 h and were detected by CCK8. Each treatment concentration was tested in sextuplicate.

**Figure 3 ijms-23-10124-f003:**
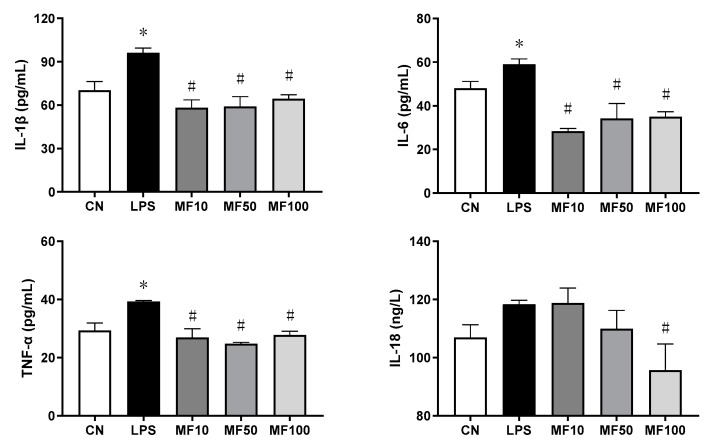
Level change of pro-inflammatory cytokines influenced by mangiferin. The cells were induced by 1 μg/mL LPS and treated with mangiferin for 24 h, and then the supernatant was detected by ELISA kits. Each group was detected in triplicate, and * means *p* < 0.05 compared with the CN group, and # means *p* < 0.05 compared with the LPS group. MF10, MF50, and MF100 mean the group treated with the concentration of 10 μg/mL, 50 μg/mL, and 100 μg/mL mangiferin, respectively.

**Figure 4 ijms-23-10124-f004:**
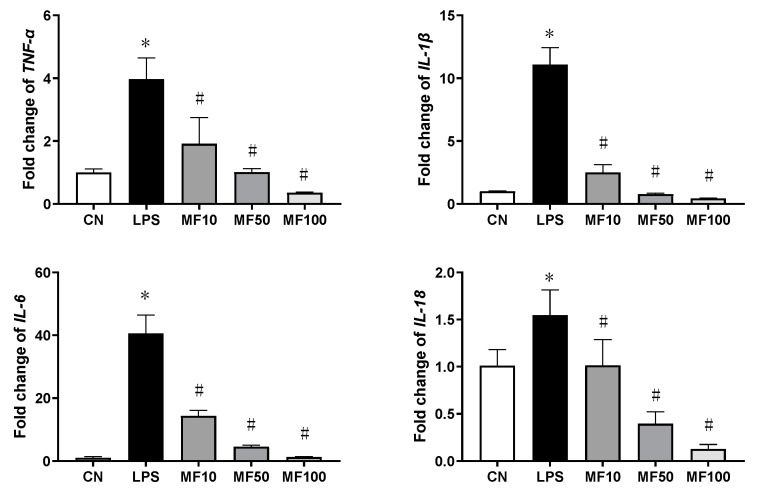
The mRNA expression changes influenced by mangiferin. The cells were induced by 1 μg/mL LPS and treated with mangiferin for 24 h, then the cells’ total mRNA was used to test the gene expression using RT-qPCR. Each group was detected in triplicates, and * means *p* < 0.05 compared with the CN group, and # means *p* < 0.05 compared with the LPS group. MF10, MF50, and MF100 mean the group treated with the concentration of 10 μg/mL, 50 μg/mL, and 100 μg/mL mangiferin, respectively.

**Figure 5 ijms-23-10124-f005:**
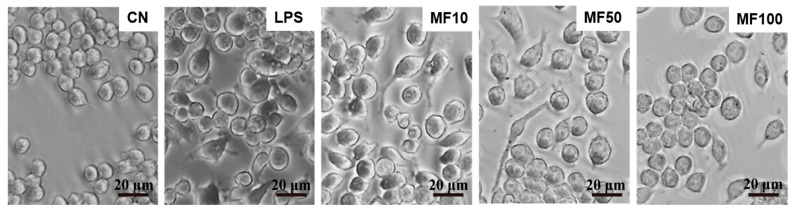
Cell pyroptotic morphology observation influenced by mangiferin. The cells were seeded in a 6-well culture plate and cultured for 24 h, and then the cells were induced by 1 μg/mL LPS and treated with different doses of mangiferin. After treatment for 24 h, cell morphology was observed under phase-contrast microscope. The scale bars in the figures represent 20 μm, and MF10, MF50, and MF100 mean the group treated with the concentration of 10 μg/mL, 50 μg/mL, and 100 μg/mL mangiferin, respectively.

**Figure 6 ijms-23-10124-f006:**
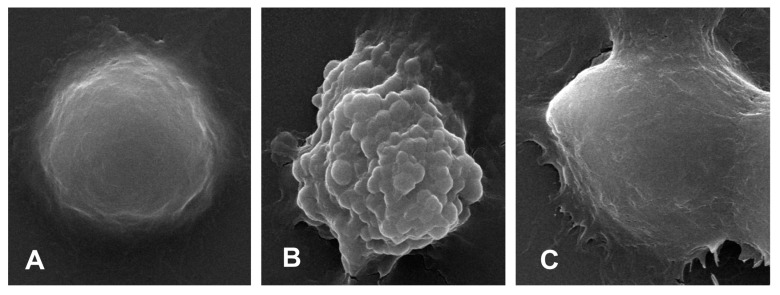
Cell ultramicroscopic morphological observation under scanning electronic microscopy. (**A**) is the negative control group, (**B**) is the pyroptotic cell induced by LPS, and (**C**) is the pyroptotic cell treated with 50 μg/mL mangiferin. The magnification times was 10,000×.

**Figure 7 ijms-23-10124-f007:**
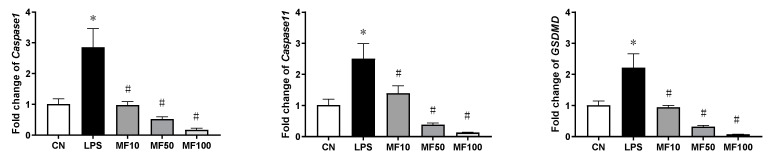
Gene expression influenced by mangiferin in BMDMs. The cells were induced by LPS and treated with mangiferin for 24 h, and then cells’ total mRNA was used to test the gene expression using RT-qPCR. Each group was detected in triplicate, and * means *p* < 0.05 compared with the CN group, and # means *p* < 0.05 compared with the LPS group. MF10, MF50, and MF100 mean the group treated with the concentration of 10 µg/mL, 50 µg/mL, and 100 µg/mL mangiferin, respectively.

**Figure 8 ijms-23-10124-f008:**
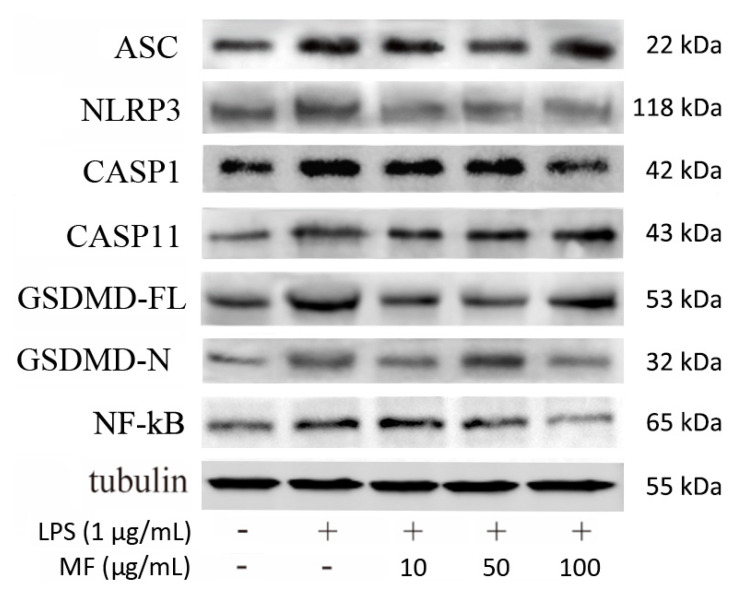
Mangiferin effected inflammatory and pyroptosis-associated protein expression in BMDMs. Each group was tested in triplicate and the protein band picture is a typical representative. In the band picture, the protein symbol ASC is apoptosis-associated speck-like protein containing a caspase-recruitment domain, NLRP3 is Nod-like receptor protein-3, CASP1 is caspase-1, CASP11 is caspase-11, GSDMD-FL is full-length gasdermin D, and GSDMD-N is N-terminal cleaved from gasdermin D.

**Figure 9 ijms-23-10124-f009:**
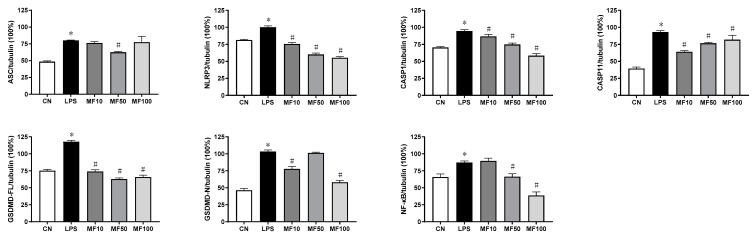
Statistical analysis of protein band gray levels. The gray bands were analyzed by Image J, and the values were described as means ± standard deviation. * means *p* < 0.05 compared with the CN group, and # means *p* < 0.05 compared with the LPS group. MF10, MF50, and MF100 mean the group treated with the concentration of 10 µg/mL, 50 µg/mL, and 100 µg/mL mangiferin, respectively.

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
