# Peer review of "Anti-Inflammation and Anti-Pyroptosis Activities of Mangiferin via Suppressing NF-κB/NLRP3/GSDMD Signaling Cascades"

_ijms, 2022, doi:10.3390/ijms231710124_

Round 1
Reviewer 1 Report
The manuscript by Feng et al describes the anti-inflammatory properties of mangiferin. While the overall approach is planned well the manuscript offers little and is essentially a repeat of other studies using this molecule.
My main comments are;
Please check for spelling, formatting, and grammatical errors that appear throughout the current manuscript.
Mangiferin is widely known to posses anti-inflammatory properties when tested in mammalian models. The current manuscript lacks novelty, could additional experiments be performed or additional data included that could add novelty to the current work? This would make the manuscript more appealing.
All figures should be formatted and made larger. Figure 1 would be better presented as a bar graph, that is supported by the use of standard deviation/standard error bars. Other figures are small and difficult to interpret.
The western blots (figure 7), are fine but ideally a loading standard should be provided for each protein target.
Minor comment.
Make greater use of the figure and table legends. This should be more descriptive and informed.
Figure 9, should be moved presented as Figure 1.
Reviewer 2 Report
There are still many problems in this manuscript, so it is suggested to further revise it.
1. It is suggested that the abstract should be written according to the requirements of int.j.mol.sci.
2. The introduction section introduces a lot of information about sepsis, but the full text is only a study of the pathway. The treatment of sepsis is only an application prospect in the conclusion section without more data support. This arrangement seems to have a big problem.
3. Figure 1 should usually be represented by column chart, and the author repeats it 6 times, but why not provide the standard error?
4. The figure 4 lacks scale bar.
5. Part of the result analysis is too simple.
6. The author gives Figure 9, but Figure 9 is neither quoted nor explained.
7. There are some spelling or formatting errors in the manuscript, such as℃, CO2, etc.
Round 2
Reviewer 1 Report
Dear authors,
It is clear that many of the comments by the reviewers have been taken onboard and amendments have been made to the text and figures. Again, in the introduction, the emphasis on sepsis is unnecessary since the paper is largely a description of the mechanisms of action of mangiferin in inflammation. To focus on sepsis would require the inclusion of an in vivo model of sepsis to support the cellular model/mechanisms used. My main concern is the lack of novelty, as this work is essentially a repeat of past papers and offers little in our understanding of the bioactive nature of mangiferin.
Author Response
Dear reviewer
Thank you very much for your comments. According to your comments, we have amended Introduction of the manuscript. Specifically, we replaced some references to introduce cell pyroptosis in vitro, because we failed to conducted experiment of an in vivo model of sepsis and treatment. Actually, there are some observation reports on protecting function against sepsis by mangiferin. Thus, we thought there is no need to repeat the experiment. However, the sophisticated mechanism and pharmacological targets of anti-pyroptosis of mangifeirin remain need further study. In this study, we demonstrated the concrete targets in signaling cascades of pyroptosis, which were regulated by mangiferin. We hold that this is the novelty of our study. Lastly, thank you very much again. Please find acceptance to our revised manuscript and agree publication.
Reviewer 2 Report
The author doesn't seem to take this revision seriously. The cover letter uploaded by the author seems to be a list of abbreviations. Therefore, I can only reserve my original opinion.
Author Response
Response to Reviewer 2 Comments
Dear reviewer,
Thank you very much for your kindly comments on our manuscript, and apologize for our carelessness of comments response uploading. The comments are valuable and very helpful to promote our paper. According to the editor’s and reviewer’s comments, we revised the manuscript again, and all of the revised content were outlined in red color. Moreover, we would like to answer the reviewer’s questions and give a detailed account of the changes made to the original manuscript as follow.
Point 1: It is suggested that the abstract should be written according to the requirements of int.j.mol.sci.
Response 1: Your suggestion is very nice. We have revised the abstract according to the journal requirements.
Point 2: The introduction section introduces a lot of information about sepsis, but the full text is only a study of the pathway. The treatment of sepsis is only an application prospect in the conclusion section without more data support. This arrangement seems to have a big problem.
Response 2: As comments of the reviewer, there are many reports that mangiferin has significant activities of anti-inflammation and protect function against sepsis, however, the mechanism of mangiferin to regulate pyroptosis and inflammation remain needs further study, because there is no the sophisticated mechanism and pharmacological target of anti-pyroptosis of mangifeirin. In this study, we demonstrated the concrete targets of mangiferin, and shed a new sight on mangiferin development. According to your suggestions, we have supplemented this opinion in the Introduction.
Point 3: Figure 1 should usually be represented by column chart, and the author repeats it 6 times, but why not provide the standard error?
Response 3: According to your suggestion, Figure 1 has been modified as column chart with standard error. Additionally, we have adjusted Figure 1 into Figure 2 in revised manuscript.
Point 4: The figure 4 lacks scale bar.
Response 4: This is an carelessness of our writing, we have modified the image and added scale bar in the picture.
Point 5: Part of the result analysis is too simple.
Response 5: In accordance with the comments of the reviewers, we provide a detailed description of each study result in revised manuscript, please re-review it again.
Point 6: The author gives Figure 9, but Figure 9 is neither quoted nor explained.
Response 6: According to the structure of the article, we have adjusted Figure 9 into Figure 1 and quoted it at the section of Introduction.
Point 7: There are some spelling or formatting errors in the manuscript, such as℃, CO2, etc.
Response 7: Thank you very much for your valuable comments. We have invited a native English researcher to review the spelling and formatting throughout of the manuscript, and have amended these errors in revised work.
Lastly, thank you again for your positive and constructive comments on our manuscript. We hope you could find acceptance to our revised manuscript and agree publication.
Round 3
Reviewer 2 Report
Thanks to the editor's patience and responsibility, it is possible for this manuscript to be further revised. At present, the author has answered my question, and I am happy to have the manuscript published.